# Copy number variation and population-specific immune genes in the model vertebrate zebrafish

**Yannick Schäfer[1], Katja Palitzsch[1], Maria Leptin[1], Andrew R Whiteley[2], Thomas Wiehe[1]\*, Jaanus Suurväli[1,3]\***

[1]Institute for Genetics, University of Cologne, Cologne, Germany; [2]WA Franke College of Forestry and Conservation, University of Montana, Missoula, United States; [3]Department of Biological Sciences, University of Manitoba, Winnipeg, Canada

**Abstract** Copy number variation in large gene families is well characterized for plant resistance genes, but similar studies are rare in animals. The zebrafish (*Danio rerio*) has hundreds of NLR immune genes, making this species ideal for studying this phenomenon. By sequencing 93 zebrafish from multiple wild and laboratory populations, we identified a total of 1513 NLRs, many more than the previously known 400. Approximately half of those are present in all wild populations, but only 4% were found in 80% or more of the individual fish. Wild fish have up to two times as many NLRs per individual and up to four times as many NLRs per population than laboratory strains. In contrast to the massive variability of gene copies, nucleotide diversity in zebrafish NLR genes is very low: around half of the copies are monomorphic and the remaining ones have very few polymorphisms, likely a signature of purifying selection.

## Editor's evaluation

This useful study employs a sequence capture approach to characterize the diversity of NLR sequences in wild zebrafish populations. The authors provide solid evidence that wild zebrafish populations harbor several thousand NLR genes in total, with individual fish having a few hundred NLR gene copies.

**\*For correspondence:** twiehe@uni-koeln.de (TW); jaanus.suurvali@gmail.com (JS)

**Competing interest:** The authors declare that no competing interests exist.

## Introduction

The innate immune system of an organism provides the first defense line against pathogens. Immune genes tend to evolve quickly and are often associated with a high degree of genetic variability. Many genes and proteins of the immune system are lineage-specific (limited to specific groups of animals, plants, or other taxa), while others have defense roles in a wide range of species. In particular, proteins containing a large nucleotide-binding domain followed by smaller repeats have an immune function in animals, plants, fungi, and bacteria alike (*Ting et al., 2008*; *Jones and Dangl, 2006*; *Uehling et al., 2017*; *Gao et al., 2022*). In animals, these repeats are usually leucine-rich repeats (LRRs) and the proteins themselves are classified as NLRs (nucleotide binding domain leucine-rich repeat containing, also known as NOD-like receptors). They have a multitude of functions: some act as pathogen sensors or transcription factors (*Almeida-da-Silva et al., 2023*), others are components or modulators of inflammasomes, large protein complexes that are assembled within cells as part of the response to biological or chemical danger (*Almeida-da-Silva et al., 2023*).

**eLife digest** Humans and other animals have immune systems that protect them from bacteria, viruses and other potentially harmful microbes. Members of a family of genes known as the NLR family play various roles in helping to recognize and destroy these microbes. Different species have varying numbers of NLR genes, for example, humans have 22 NLRs, but fish can have hundreds. 400 have been found in the small tropical zebrafish, also known as zebra danios.

Zebrafish are commonly used as model animals in research studies because they reproduce quickly and are easy to keep in fish tanks. Much of what we know about fish biology comes from studying strains of those laboratory zebrafish, including the 400 NLRs found in a specific laboratory strain. Many NLRs in zebrafish are extremely similar, suggesting that they have only evolved fairly recently through gene duplication. It remains unclear why laboratory zebrafish have so many almost identical NLRs, or if wild zebrafish also have lots of these genes.

To find out more, Schäfer et al. sequenced the DNA of NLRs from almost 100 zebrafish from multiple wild and laboratory populations. The approach identified over 1,500 different NLR genes, most of which, were previously unknown. Computational modelling suggested that each wild population of zebrafish may harbour up to around 2,000 NLR genes, but laboratory strains had much fewer NLRs. The numbers of NLR genes in individual zebrafish varied greatly – only 4% of the genes were present in 80% or more of the fish. Many genes were only found in specific populations or single individuals.

Together, these findings suggest that the NLR family has expanded in zebrafish as part of an ongoing evolutionary process that benefits the immune system of the fish. Similar trends have also been observed in the NLR genes of plants, indicating there may be an evolutionary strategy across all living things to continuously diversify large families of genes. Additionally, this work highlights the lack of diversity in the genes of laboratory animals compared with those of their wild relatives, which may impact how results from laboratory studies are used to inform conservation efforts or are interpreted in the context of human health.

Plants have their own NLRs that are structurally similar to the ones from animals and also carry out central functions in the immune response (*Urbach and Ausubel, 2017*; *Yue et al., 2012*). Their diversity has been extensively characterized in several species, including the thale cress (*Arabidopsis thaliana*), and vastly different repertoires have been found from different strains or individuals (*Van de Weyer et al., 2019b*). NLR repertoires can also be referred to as NLRomes, and a species-wide repertoire is called the 'pan-NLRome'.

Most knowledge about NLRs in animals comes from studies of humans and rodents, but their NLR repertoires (20–30 genes) are smaller than those of many other species such as the purple sea urchin, the sponge *Amphimedon queenslandica*, and many fish (*Hibino et al., 2006*; *Yuen et al., 2014*; *Suurväli et al., 2022*). However, even in mice one NLR (*Nlrp1*) has different copy numbers in different laboratory strains, ranging from 2 to 5 (*Lilue et al., 2018*). In many fishes, studies have reported NLR repertoires in the range of 10–50 genes (e.g., *Rajendran et al., 2012*; *Li et al., 2016*). In others, hundreds of NLRs are present, including in the model species zebrafish (*Danio rerio*) (*Stein et al., 2007*; *Laing et al., 2008*; *Tørresen et al., 2018*; *Adrian-Kalchhauser et al., 2020*; *Suurväli et al., 2022*). The zebrafish reference genome contains nearly 400 NLR genes, two-thirds of which are located on the putative sex chromosome (chromosome 4), in a genomic region associated with extensive haplotypic variation (*Howe et al., 2013*; *Howe et al., 2016*; *McConnell et al., 2023*; *Anderson et al., 2012*).

The majority of fish NLRs represent a fish-specific subtype that was originally labeled NLR-C (*Laing et al., 2008*), although they can be further divided into at least six groups based on structural similarities and sequence of conserved exons (*Howe et al., 2016*; *Adrian-Kalchhauser et al., 2020*). A schematic structure of proteins encoded by zebrafish NLR-C genes is presented in *Figure 1A*. All of them possess a FISNA domain (fish-specific NACHT-associated domain), which precedes the nucleotide-binding domain NACHT and is encoded by the same large exon near the N-terminus of the protein (*Howe et al., 2016*). FISNA-NACHT is in some cases preceded by the effector domain PYD, but this is encoded by a separate exon (*Howe et al., 2016*). Additionally, many NLR-C proteins have a B30.2

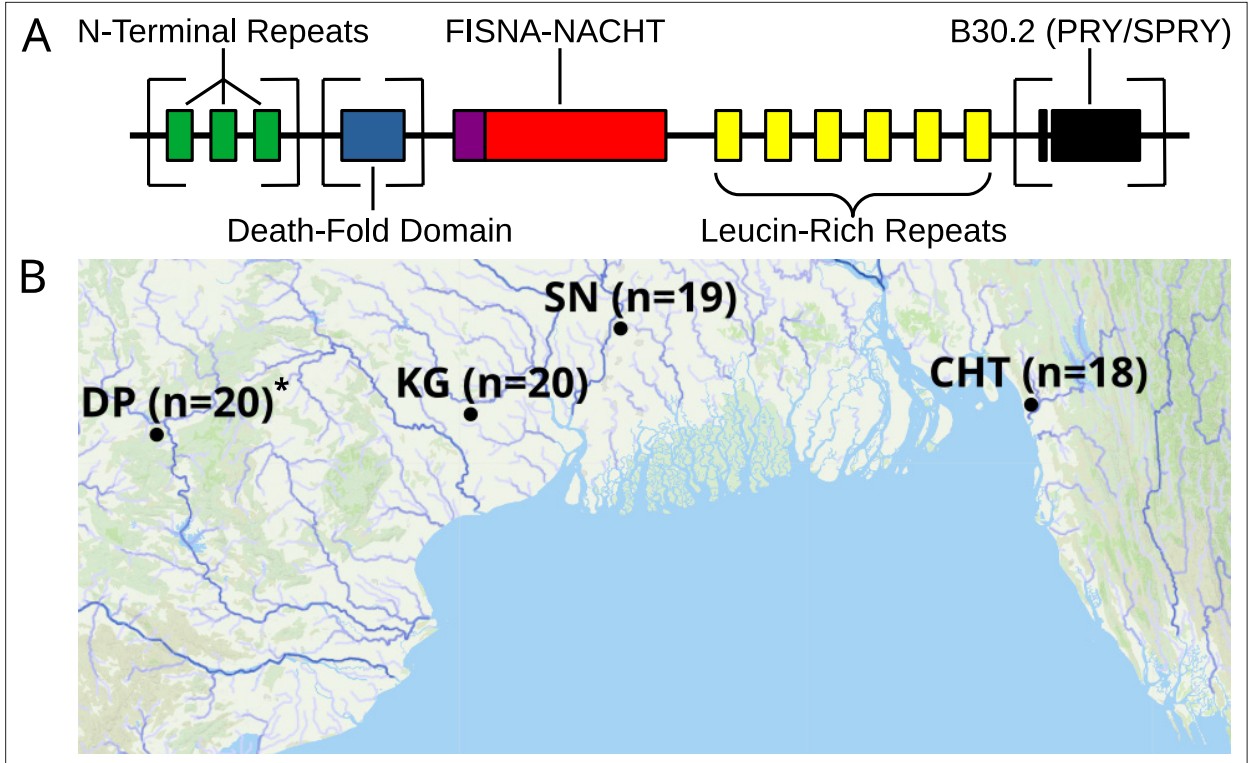

**Figure 1.** Structure of zebrafish NLRs and a map showing the origin of wild zebrafish samples. (**A**) Generalized, schematic representation of the domain architecture of an NLR-C protein. Each box represents a translated exon. The N-terminal repeats, the death-fold domain, as well as the B30.2 domain only occur in subsets of NLR-C genes. The number of N-terminal repeats and leucine-rich repeats can vary. Domains that can be either present or absent in different NLRs are surrounded by square brackets. (**B**) Sampling sites for wild zebrafish. All sites are located near the Bay of Bengal. Final sequenced sample sizes are indicated in parentheses. The map is based on geographic data collected and published by AQUASTAT from the Food and Agriculture Organization of the United Nations (**FAO, 2021**). The population DP is marked with an asterisk because its analysis and results are presented only in figure supplements.

domain (also known as PRY/SPRY) at the C-terminal end, separated from FISNA-NACHT by multiple introns and exons containing the LRRs (*Figure 1A*; *Howe et al., 2016*). The B30.2 domain functions through protein–protein interaction (*Woo et al., 2006*) and is also found in a variety of other genes such as the large family of TRIM ubiquitin ligases (*van der Aa et al., 2009*; *Howe et al., 2016*; *Suurväli et al., 2022*) that are often also involved in immunity.

It is not known why fishes possess so many NLRs, how they evolve, and how much within-species genetic variability they have. The previously observed repeated expansions and contractions of this family suggest it to have a high rate of gene birth and death (*Suurväli et al., 2022*). Studies have shown that viral and bacterial infections can induce the expression of specific fish NLRs (reviewed in *Chuphal et al., 2022*). Some of these have PYD or CARD domains and can even form inflammasomes similar to mammalian NLRs (*Kuri et al., 2017*; *Li et al., 2018b*). A species-wide inventory of major NLR exons in a model species such as zebrafish would provide valuable insights into the evolution and diversity of this large immune gene family.

## Results

By adapting and modifying a protocol that combines bait-based exon capture with PacBio SMRT technology (*Witek et al., 2016*), we successfully generated circular consensus sequence (CCS) data for targeted parts of the immune repertoire from 93 zebrafish (of initial 96), representing four wild populations (*Figure 1B*) and two laboratory strains. With this approach, we aimed to sequence all exons in zebrafish that encode the nucleotide-binding FISNA-NACHT domains and all exons that encode B30.2 domains. Samples of one wild population (DP) suffered from poor sequence coverage and had

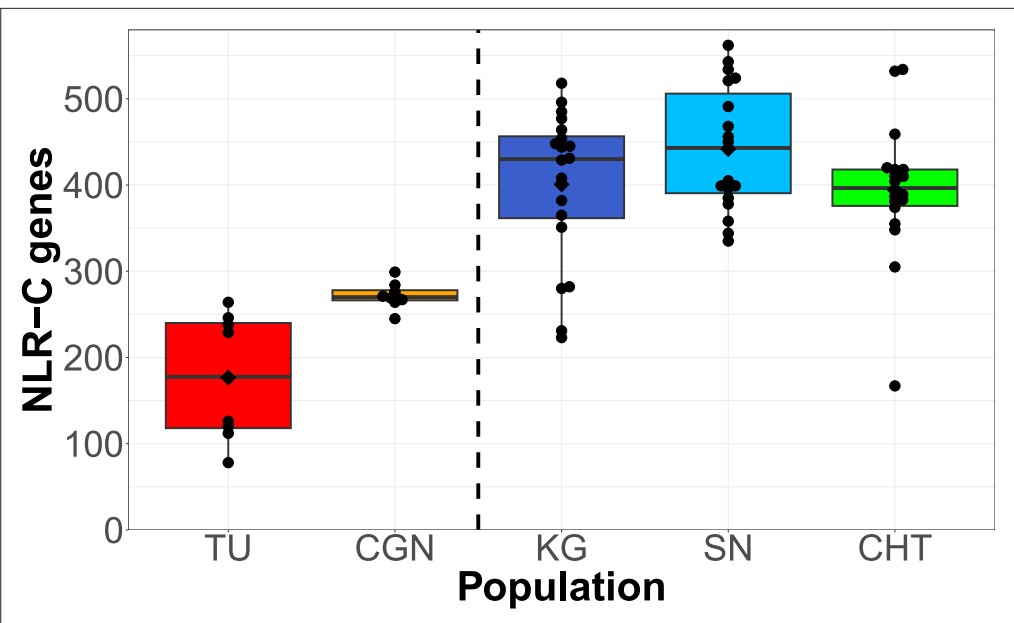

**Figure 2.** Total counts of NLRs found per individual, shown for each population. Black diamonds on the box plots denote means, horizontal lines denote medians. Left side: two laboratory strains; right side: three wild populations.

The online version of this article includes the following source data and figure supplement(s) for figure 2:

**Source data 1.** Source tables for *Figure 2* and its supplements.

**Source data 2.** Sequences and target locations of RNA baits.

**Figure supplement 1.** Sequencing and assembly statistics of circular consensus sequence (CCS) reads from NLR exons.

**Figure supplement 2.** Assembled NLRs in the reference genome GRCz11.

**Figure supplement 3.** Identification of B30.2 domains associated with zebrafish NLRs.

to be excluded from downstream analyses in order to avoid bias in interpretation. Results involving this population are only shown in figure supplements and not in the main figures.

Our protocol used PCR with primers targeting ligated adapters to amplify the below-nanogram amounts of genomic DNA obtained from exon capture. This limited our fragment sizes to the lengths of what the polymerase was able to amplify. Zebrafish NLRs can have their exons spread out across tens of kilobases, so that we cannot know which exons belong to the same gene. However, we were able to use captured sequence surrounding the targeted exons to distinguish among near-identical coding sequences and separate NLR-associated B30.2 domains from B30.2 elsewhere in the genome.

### The zebrafish pan-NLRome

We used an orthology clustering approach on NLR sequences assembled from all populations to create a reference set of NLRs (a pan-NLRome). This resulted in the identification of 1513 unique FISNA-NACHT containing sequences and 567 for NLR-associated B30.2 (NLR-B30.2). Nearly 10% of the sequences (145 FISNA-NACHT and 64 NLR-B30.2) contained pre-mature stop codons that were at least 10 amino acids from the end and led to early truncation of the protein. In total, 101 of the 1513 FISNA-NACHT were preceded by an exon containing the N-terminal effector domain PYD. Nearly all of those (97 out of 101) were found in group 1 NLR-C genes identified by the presence of the characteristic sequence motif GIAGVGKT (*Howe et al., 2016*). Since the combination of FISNA and NACHT is only present in NLR-C, its count of 1513 can be considered equal to the total number of NLR-C genes in the data. We found each individual zebrafish to have 100–550 NLR genes from the pan-NLRome in at least one copy (*Figures 2 and 3*), and only 50–75% of these have a high-quality match in the GRCz11 reference genome (*Figure 2—figure supplement 2*). In general, laboratory zebrafish had less NLRs than wild samples (*Figure 2*). The number and length of CCS reads and assembled contigs (both prior to orthology clustering) are presented in *Figure 2—figure supplement 1*.

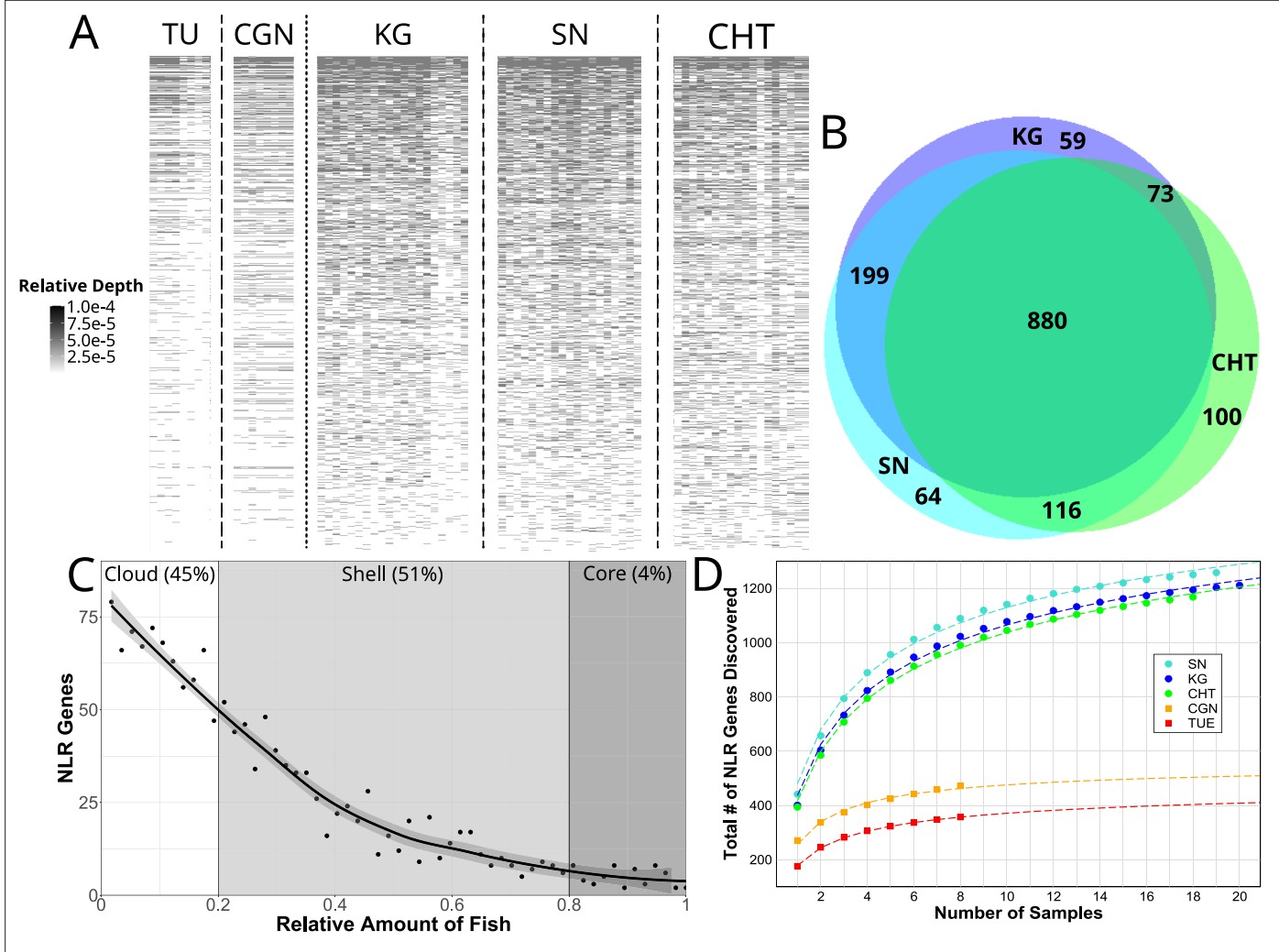

**Figure 3.** Copy number variation of NLR genes. (**A**) Sequence data from each individual zebrafish (vertical axis) was aligned to FISNA-NACHT exon sequences of the pan-NLRome (horizontal axis). Grayscale intensity shows, for each NLR, the proportion of NLR-aligning data in each given fish that matches this specific gene. Darker gray indicates a higher likelihood of this NLR being represented in multiple copies in the particular individual. Light gray indicates a single copy, white indicates absence. For clarity, only the 1235 FISNA-NACHT exons for which at least one fish had a minimum of 10 reads mapped to it are shown. (**B**) Numbers of pan-NLRome sequences (based on FISNACHT diagnosis) found in all three, two, or only one wild population. (**C**) Relative numbers of fish in which pan-NLRome sequences were found in wild populations. 'Core' pan-NLRome: genes which are found in at least 80% of the sample (from a total of 57 wild fish); 'shell': genes in at least 20%; 'cloud': rare genes found in less than 20% of the sample. (**D**) Observed and estimated sizes of population-specific pan-NLRomes. Data points (filled circles and squares) show the average number of totally discovered NLR genes (as identified via their FISNA-NACHT domain) when investigating $x$ fish. The dashed line is obtained by non-linear fit of the data to the function given in *Equation 2*. For all populations, the hypothetical pan-NLRome size – when extrapolating $x \to \infty$ – is finite (see *Table 1*).

The online version of this article includes the following source data and figure supplement(s) for figure 3:

**Source data 1.** Source tables for *Figure 3* and its supplements.

**Figure supplement 1.** Comparison of copy number variation in FISNA-NACHT and NLR-B30.2 exons.

**Figure supplement 2.** Copy number variation of NLR genes, including the DP population.

Whereas FISNA-NACHT is only found in NLRs, B30.2 domains are also found in other gene families. In addition to the 567 NLR-B30.2 domains, we also found 732 B30.2 domains not associated with NLRs. We were able to distinguish between them by utilizing the sequence of a short highly conserved 47 bp exon that appears to precede B30.2 in NLRs, but not in other genes (*Figure 2— figure supplement 3*). Each individual zebrafish possesses 20–180 NLR-B30.2s n at least one copy (*Figure 3—figure supplement 1*).

**Table 1.** Values of fitted parameters and saturation limits for FISNA-NACHT and NLR-B30.2 exons, by population.

| Population | FISNA-NACHT | | | | NLR-B30.2 | | | |
|---|---|---|---|---|---|---|---|---|
| - | α | β | Limit | Quantile* | α | β | Limit | Quantile* |
| TU | 178.274 | 1.43356 | 519.548 | 118 | 53.8579 | 1.40774 | 164.73 | 164 |
| CGN | 257.207 | 1.62786 | 569.367 | 23 | 78.7156 | 1.61283 | 177.246 | 25 |
| DP | 309.14 | 1.01231 | 25284 | 2930† | 69.3609 | 0.87454 | ∞ | na |
| KG | 436.761 | 1.2152 | 2288.41 | 2060 | 145.715 | 1.1418 | 1113.23 | 6.41e6 |
| SN | 479.892 | 1.26093 | 2152.12 | 3907 | 145.548 | 1.10183 | 1514.35 | 3.75e9 |
| CHT | 416.712 | 1.18893 | 2451.81 | 1.12e5 | 135.677 | 1.11911 | 1218.54 | 1.41e8 |

*Sample size required to capture 90% of the population's pan-NLRome.
†DP required sample size refers to only 10% (instead of 90%) of its hypothetical pan-NLRome size.

## Copy number variation in the pan-NLRome

Aligning CCS reads to the pan-NLRome revealed a considerable amount of variability in the proportion of reads mapping to them, both between and within populations (*Figure 3A*). This can be interpreted as the gene being present in different copy numbers. Furthermore, each NLR had its own distinct pattern of copy number variation, although generally the highest copy numbers were observed for the wild populations KG, SN, and CHT (*Figure 3A*). We also observed some sequencing batch-related differences, but the copy numbers differed even between individuals sequenced in the same batch.

Of the 1513 unique FISNA-NACHT and 567 NLR-B30.2 sequences, 880 FISNA-NACHT and 346 NLR-B30.2 (59 and 57%, respectively) were detected in at least one individual from all wild populations (*Figure 3B*, *Figure 3—figure supplement 1*).

There were also NLR sequences shared between just two wild populations, and some were restricted to a single population (*Figure 3B*). Moreover, we observed a lot of variability in the distributions of gene copies among fish within populations (*Figure 3C*). Only around 4% of the genes in the pan-NLRome were found in 80%, or more, of the wild fish. They constitute the core NLRome (*Van de Weyer et al., 2019a*). Most genes (51%) were found in the so-called shell of the pan-NLRome (20–80% of fish). Almost as many (45%) are found in a few fish (less than 20% of the sample) only. Although 60% of NLR genes occur in all wild populations, only 4% are omnipresent, that is, are in the core pan-NLRome. Thus, there is considerable variation in the NLR repertoires of individuals from the same population.

The total number of NLRs identified in a number $x$ of individual fish can be fitted to a harmonic function (*Medini et al., 2020*). Using this function (see 'Materials and methods'), we estimated the sizes of the NLRomes of the populations (*Figure 3D*) and found a total of 520 and 570 NLRs in the laboratory strains TU and CGN, respectively (*Table 1*). For the wild populations, we estimated four times as many: 2283 in KG, ,896 in SN, and 2452 in CHT.

## Differences from the reference genome

NLRs sequenced in this study were often different from those present in the reference genome GRCz11. Even NLRs sequenced from the strain that the reference genome itself is based on (TU) did not always align well to it. When the exon itself did align, the intronic sequences surrounding it could often be very different from the reference. In numbers, only around 75% of NLRs occurring in TU fish aligned to the reference genome GRCz11 with high mapping qualities (*Figure 2—figure supplement 2A*). This number dropped even lower elsewhere – from 60–65% of NLRs in CGN which aligned well to the reference, down to only around 50% for the wild populations. The majority of NLRs that did not map well had a very poor mapping quality of 1 (*Figure 2—figure supplement 2B*). Moreover, there were 9 FISNA-NACHT and 10 NLR-associated B30.2 in the pan-NLRome which did not map anywhere in the reference genome.

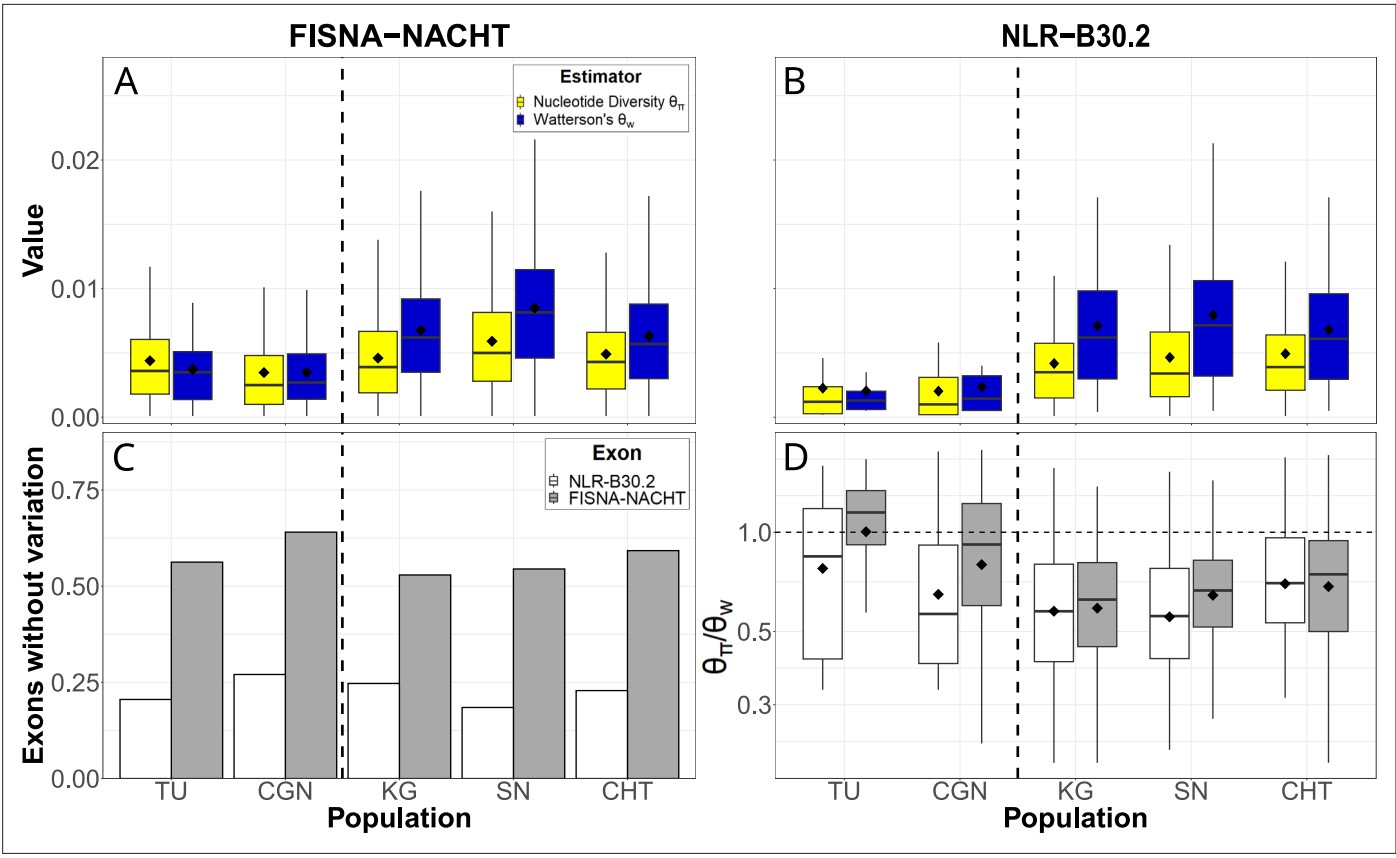

**Figure 4.** Single-nucleotide variation in NLR exons. Pairwise nucleotide diversity ($\theta_\pi$) and Watterson's estimator of the scaled mutation rate ($\theta_w$) for FISNA-NACHT (**A**) and NLR-associated B30.2 (**B**) exons. (**C**) Proportion of exons without any single nucleotide polymorphisms. (**D**) Ratio of $\theta_\pi/\theta_w$. Only exons with at least one single-nucleotide polymorphism are shown. The dotted, horizontal line marks a ratio of 1, the expected value under neutrality and constant population size. The black diamonds on box plots denote means, horizontal lines denote medians.

The online version of this article includes the following source data and figure supplement(s) for figure 4:

**Source data 1.** Source tables for *Figure 4* and its supplement.

**Figure supplement 1.** Single-nucleotide polymorphisms of different NLR exons shown by population, including DP.

### Purifying selection on single-nucleotide variants

We used the pan-NLRome as a reference for identifying single-nucleotide polymorphisms in the data. NLR sequence diversity was rare, with a large fraction of exons not having any variants in any of the populations. If variants were present, nucleotide diversity ($\theta_\pi$) was up to 0.016 and Watterson's estimator ($\theta_w$) up to 0.021 (*Figure 4A and B*). In laboratory strains, genetic variability of FISNA-NACHT exceeded that of B30.2, but no such pattern was observed for wild populations. B30.2 exons of laboratory strains were also less variable than B30.2 from wild zebrafish (*Figure 4B*). The proportion of exons without any polymorphisms was much higher among FISNA-NACHT than among B30.2 (*Figure 4C*). The majority of variable NLR exons had $\theta_\pi/\theta_w$ ratios of less than 1 (*Figure 4D*), indicating an excess of rare alleles.

### Discussion

We sequenced and assembled the FISNA-NACHT and B30.2 exons of hundreds of NLRs from 93 zebrafish. We were able to capture the diversity of this gene family in three wild populations and two laboratory strains, and produced lower coverage NLR data for an additional wild population (DP). Analyzing the 73 zebrafish from populations other than DP, we found evidence that each genome from a wild individual contains only a fraction of more than 1500 identified NLR copies. The number of NLRs found per individual, each with one or more copies, ranged from around 100–550. Some of

the lower counts were likely underestimated due to low sequencing depths in specific samples. Since all samples from population DP suffered from low read depth, their analysis is only shown in figure supplements. As targeted sequencing based on bait-capture requires sufficient homology between bait and target, diverged NLR exons may have been missed in our approach. This affects B30.2 exons even more than FISNA-NACHT exons because they are much shorter. However, the observed slow increase in newfound NLR gene copies per sequenced individual after the first few individuals indicates that not many NLRs were missed. The sizes of NLR repertoires differ between zebrafish individuals in the three wild populations.

Nonlinear fitting of NLR counts to *Equation 2* suggested that the investigated populations all possess closed pan-NLRomes with roughly 500–600 NLRs in the laboratory strains and around 2000 NLRs in the wild populations. The total numbers of NLRs with a B30.2 exon are about 170 in the laboratory strains and between 1100 and 1500 for the wild populations (*Table 1*). To explore the entire NLRome of wild populations, large samples are needed: based on the curve-fitting results, we estimate that capturing 90% of the NLRome may require up to several hundred thousand fish (*Table 1*). Orthogroup clustering with the data from DP resulted in 47 FISNA-NACHT exons which did not occur in any other population. Our results suggest that the pan-NLRome of the entire species must be vastly larger than what we have been able to detect with our limited sample sizes from a limited number of populations. Geographically distant populations – for example, in Nepal or the Western Ghats (*Whiteley et al., 2011*) – likely harbor many more NLRs which are not present in the populations we sequenced.

Although a few zebrafish assemblies are available in addition to the reference genome, for instance, the fDanRer4.1 assembly from the Tree of Life Initiative (GCA_944039275.1), none of those provide a suitable framework for mapping and analyzing NLRs on their own. One of the hindrances is the fact that the majority of NLR genes are located on the notoriously difficult to assemble long arm of chromosome 4, which harbors plenty of structural variation (*McConnell et al., 2023*). Furthermore, large multi-copy gene families are difficult to analyze. Read mapping and counting of copies in a particular genome is not trivial. Any downstream analysis which relies on clearly distinguishing paralogous and orthologous comparisons becomes fuzzy, if not impossible. Still, improving sequencing technology and the rising interest in pan-genomic studies *Bayer et al., 2020*; *Sherman and Salzberg, 2020*; *Liao et al., 2023* have already started to transform the data structures in which genomes are stored, away from a single-reference genome-based view, toward graph-based genome networks. Whether the promise of a thereby improved inventory of structural variation of a species holds up remains still to be seen. Anyway, as shown for the zebrafish NLRs, the availability of a single high-quality reference genome is certainly not sufficient neither to identify nor to understand the diversity of large gene families.

## Properties of the zebrafish NLRome

We have previously demonstrated a substantial reduction in single-nucleotide variation in zebrafish laboratory strains compared to wild populations (*Suurväli et al., 2020*). Here, we showed that the copy numbers of the NLRome and their variation are also heavily reduced. The most obvious explanation for this observation is the recent population bottleneck which marks the establishment of laboratory strains. The reduction in copy number variation in the major histocompatibility complex (MHC) locus in a population of greater prairie-chicken was attributed to a recent bottleneck as well (*Eimes et al., 2011*). Additionally, the reduced amount of pathogenic challenges in a laboratory environment could lead to a steady loss of expendable genes. For these reasons, one has to exercise caution when extending conclusions from immune-related studies on laboratory zebrafish to wild zebrafish. The same caution should also be exercised when extending results from laboratory organisms to other species, including human.

Studies have shown that even mammals have hundreds of genes with diverse molecular functions that are affected by copy number variation, even though it rarely involves full genes (*Kooverjee et al., 2023*; *Zarrei et al., 2015*). One example of the latter is the MHC locus, which harbors varying numbers of gene copies between closely related species of ruminants (*He et al., 2024*) and has haplotype-specific copies in mice (*Lilue et al., 2018*). However, the vast number of NLRs in zebrafish combined with presence/absence variation (*McConnell et al., 2023*) and high rates of duplication exceeds what has been found in other animals so far. A comparable situation can be found in the

NLR genes of the thale cress (*A. thaliana*). Our predicted number of NLRs in a zebrafish population is on the same scale as the 2127 NLRs found in the thale cress NLRome (*Van de Weyer et al., 2019b*). Moreover, copy numbers also vary greatly between *A. thaliana* accessions (*Lee and Chae, 2020*). A total of 464 conserved, high-confidence orthogroups were identified in *A. thaliana*, 106–143 of which were defined as the core NLRome because they were found in a subset comprising at least 80% of the accessions (*Van de Weyer et al., 2019b*). In wild zebrafish, we found a set of 880 NLR genes which were detected in at least one individual from three wild populations, but only 58 NLRs were found in the vast majority (more than 80%) of wild individuals. Although structural similarities of NLRs in plants and animals are thought to be the result of convergent evolution (*Yue et al., 2012*), it appears that the similarities extend to their evolutionary trajectories. However, the overall number of gene copies as well as the variation in copy numbers within populations and in individual gene repertoires are more extreme in zebrafish than in *A. thaliana*.

We postulate that as immune genes, many NLR genes are likely shared between populations because they provide a fitness advantage in the defense against common pathogens. The additional NLRs shared among only some of the wild populations and the population-specific NLRs may represent local adaptations to ecological niches. Additionally, there could be functional redundancy within the NLRome, so that different individuals have different NLRs with the same functional role. In general, the fact that hundreds of NLR gene copies are maintained in zebrafish, together with a signature of purifying selection, suggests that the evolution of these genes is far from neutral. Although the expression of fish NLRs is often induced by pathogen exposure (reviewed in *Chuphal et al., 2022*), the exact function of most zebrafish NLR-C genes remains unclear. It is possible that some of them participate in the formation and activity of inflammasomes (*Li, 2018a*; *Valera-Pérez et al., 2019*; *Lozano-Gil et al., 2022*, *Kuri et al., 2017*), but we only found the N-terminal effector domains (CARD or PYD) that are typically involved in this function (*Petrilli et al., 2005*) in a small subset of NLR-C genes.

Although we mainly used the counts of FISNA-NACHT orthogroups to estimate total numbers of NLRs, we also analyzed the B30.2 exons of NLR-C genes. In general, NLR-associated B30.2 exons exhibit patterns of copy number variation that are similar to those seen for FISNA-NACHT. For example, about half of the B30.2 sequences are found in all wild populations, similar to the set of 880 FISNA-NACHT exon sequences conserved among populations.

## What drives the copy number differences?

There are at least two mechanisms which could contribute to the extensive copy number variation seen among zebrafish populations: first, it could be attributed to a high degree of haplotypic variation. Large DNA fragments contain different sets of genes and gene copies, similar to the zebrafish MHC loci (*McConnell et al., 2014*). Extensive haplotypic variation occurs on the long arm of chromosome 4, the location containing over two-thirds of all NLRs in zebrafish (*McConnell et al., 2023*). Such segregating haplotype blocks would explain the existence of the core NLRome, but not the frequent presence of genes that occur only in a single individual.

Alternatively or additionally, the evolution of NLR-C genes could be driven by duplication events (*Cannon et al., 2004*) and gene conversion (*Laing et al., 2008*). Gene duplications can be caused by unequal recombination, transposon activity, or whole genome/chromosome duplications (*Magadum et al., 2013*; *Kapitonov and Jurka, 2007*). The arrangement of NLR-B30.2 genes in clusters on the long arm of chromosome 4 suggests that tandem duplication via unequal crossing-over (*Otto et al., 2022*) played the most important role in the expansion. Since there are many transposable elements on the long arm of chromosome 4 (*Howe et al., 2013*), it would be reasonable to assume that at least some of them have assisted in the local expansion and transfer of NLR exons and genes to chromosomes other than chromosome 4. Since our targeted sequencing approach does not elucidate the genomic arrangement of the NLR gene copies and many of them do not have recognizable orthologs in the reference genome, we cannot draw further conclusions about the role of tandem arrays in their evolutionary trajectory.

It is tempting to speculate that chromosome 4 could be a source of NLRs which continuously generates new copies. However, gene gains must be balanced by gene loss to maintain a stable genome size. NLR-C genes may be lost via accumulation of random mutations due to a lack of selective pressure and loss-of-function mutations, but they may also be lost through unequal recombination. This

mechanism would allow only NLR genes contributing to the functionality of the immune system to be kept, while others would disappear.

In the similarly evolving plant NLRs, tandem duplication is thought to be the primary driver of NLR gene expansion (*Cannon et al., 2004*), but they are also often associated with transposable elements. If the diversity of unrelated NLR genes in such distantly related species is driven by common molecular mechanisms, then the same mechanisms might also act on NLRs of other phylogenetic clades and even on unrelated large gene families, such as odorant receptors (*Mombaerts, 1999*).

## Conclusion

This study showcases an example of the evolutionary dynamics affecting very large gene families. The sheer amount of copy number variation that appears to be present in a single gene family of zebrafish is staggering, with different individuals each having numerous genes that are not present in all others. This can only be caused by diversity-generating mechanisms that are active even now. In this study, we have laid the groundwork for future studies investigating the molecular basis and evolutionary mechanisms contributing to the diversity of large, vertebrate gene families.

# Materials and methods

**Key resources table**

| Reagent type (species) or resource | Designation | Source or reference | Identifiers | Additional information |
|---|---|---|---|---|
| Strain (*Danio rerio*) | Cologne zebrafish; CGN; KOLN | Other | | 8 Cologne fish, AG Hammerschmidt, University of Cologne |
| Strain (*D. rerio*) | Tübingen zebrafish; TU | Other | | 8 Tübingen fish, AG Hammerschmidt, University of Cologne |
| Biological sample (*D. rerio*) | DP | Other | | 20 wild fish, Dandiapalli, India (22.22155, 84.79430) |
| Biological sample (*D. rerio*) | CHT | Other | | 20 wild fish, Chittagong, Bangladesh (22.47400, 91.78300) |
| Biological sample (*D. rerio*) | KG | Other | | 20 wild fish, Leturakhal, India (22.26189 87.27881) |
| Biological sample (*D. rerio*) | SN | Other | | 20 wild fish, Santoshpur, India (22.93765 88.55311) |
| Sequence-based reagent | Baits; RNA baits; hybridization baits | Daicel Arbor Biosciences | Cat# Mybaits-1-24 | Sequences available in *Figure 2—source data 2* |
| Commercial assay or kit | MagAttract HMW DNA Kit | QIAGEN | Cat# 67563 | |
| Commercial assay or kit | NucleoSpin Tissue Kit | MACHEREY-NAGEL | Cat# 740952.50 | |
| Commercial assay or kit | NEBNext Ultra II DNA Library Prep Kit | New England Biolabs | Cat# E7645L | |
| Sequence-based reagent | NEBNext Multiplex Oligos for Illumina | New England Biolabs | Cat# E7335L | Index Primers Set 1 |
| Commercial assay or kit | Kapa HiFi Hotstart Readymix | Kapa Biosystems | Cat# 07958935001 | |
| Commercial assay or kit | PreCR Repair Mix | New England Biolabs | Cat# M0309L | |
| Commercial assay or kit | SMRTbell Template Prep Kit 1.0-SPv3 | Pacific Biosciences | Cat# 100-991-900 | |
| Other | GRCz11 | NCBI RefSeq | RefSeq:GCF_000002035.6 | Zebrafish reference genome |

*Continued on next page*

*Continued*

| Reagent type (species) or resource | Designation | Source or reference | Identifiers | Additional information |
|---|---|---|---|---|
| Other | M220 miniTUBE, Red | Covaris | Cat# 4482266 | Used to shear DNA on Covaris ultrasonicator |
| Other | DB MyOne Streptavidin C1 | Thermo Fisher Scientific | Cat# 65001 | Used to retrieve bait-bound DNA fragments |
| Other | AMPure XP | Beckman Coulter | Cat# A63881 | Size selection beads |
| Other | Ampure PB | Pacific Biosciences | Cat# 100-265-900 | PacBio-compatible size selection beads |
| Software, algorithm | lima | Pacific Biosciences | lima:v1.0.0; lima:v1.8.0; lima:v1.9.0; lima:v1.11.0 | |
| Software, algorithm | ccs | Pacific Biosciences | ccs:v4.2.0 | |
| Software, algorithm | pbmarkdup | Pacific Biosciences | pbmarkdup:v1.0.0 | |
| Software, algorithm | pbmm2 | Pacific Biosciences | pbmm2:v1.3.0 | |
| Software, algorithm | samtools | https://doi.org/10.1093/bioinformatics/btp352 | samtools:v1.7 | |
| Software, algorithm | EMBOSS | https://doi.org/10.1016/s0168-9525(00)02024-2 | EMBOSS:v6.6.0.0 | |
| Software, algorithm | HMMER | https://doi.org/10.1093/bioinformatics/btt403 | HMMER:v3.2.1 | |
| Software, algorithm | blastn | https://doi.org/10.1186/1471-2105-10-421 | blastn:v2.11.0+ | |
| Software, algorithm | hifiasm | https://doi.org/10.1038/s41592-020-01056-5 | hifiasm:v0.15.4-r347 | |
| Software, algorithm | get_homologues | https://doi.org/10.1128/AEM.02411-13 | get_homologues:x86_64–20220516 | |
| Software, algorithm | deepvariant | https://doi.org/10.1038/nbt.4235 | deepvariant:r1.0 | |
| Software, algorithm | GLnexus | https://doi.org/10.1101/343970 | Glnexus:v1.2.7–0-g0e74fc4 | |
| Software, algorithm | vcftools | https://doi.org/10.1093/bioinformatics/btr330 | vcftools:v0.1.16 | |

## Samples

Wild zebrafish from four sites in India and Bangladesh (*Figure 1B*) had been collected in the frame of other projects (e.g., *Whiteley et al., 2011*; *Shelton et al., 2020*). Laboratory zebrafish from the Tübingen (TU) and Cologne (CGN) strains were provided by Dr. Cornelia Stein from the Hammerschmidt laboratory (Institute for Zoology, University of Cologne). All samples were stored in 95% ethanol until use. Tail fins from 20 fish per wild population and 8 fish per laboratory strain were used as starting material for the subsequent steps.

## DNA extraction, exon capture, and sequencing

Genomic DNA was extracted with kits from QIAGEN (MagAttract HMW kit) and MACHEREY-NAGEL (Nucleospin Tissue Kit), followed by shearing with red miniTUBEs on the Covaris M220 ultrasonicator. Nicks in the DNA were repaired with PreCR Repair Mix (New England Biolabs). Samples were barcoded with the NEBNext Ultra II DNA Library Prep Kit, then pooled together in batches of four or eight (details provided in Appendix 1). RNA baits for the exon capture (Daicel Arbor Biosciences) were custom-designed to target immune genes of interest (mainly NLRs, but also some others) based on version GRCz10 of the reference genome. Bait sequences and target locations are available in *Figure 2—source data 2*. Exon capture and PacBio library preparation were both done according to a protocol adapted from *Witek et al., 2016*. Libraries were sequenced at the Max Planck-Genome-Centre Cologne, with PacBio Sequel and Sequel II. Additional details are provided in Appendix 1.

## Read processing, mapping, and clustering

Raw sequences were de-multiplexed with lima. Consensus sequences of DNA fragments with at least three passes (CCS reads) were inferred with ccs, followed by PCR duplicate removal with pbmarkdup. All read mapping was done with pbmm2 (v.1.3.0), a PacBio wrapper for minimap2 (*Li, 2018a*). lima, ccs, pbmarkdup, and pbmm2 were all provided by Pacific Biosciences. Mapped files were processed and filtered with samtools (v1.7) (*Li et al., 2009*). De novo assemblies were generated with hifiasm (v0.15.4-r347) (*Cheng et al., 2021*). Tools from the HMMER suite (v3.2.1) (*Wheeler and Eddy, 2013*) were used to detect the presence of NLR-associated sequences. Contigs containing FISNA-NACHT or B30.2 were sorted into orthoclusters using get_homologues (build x86 64–20220516) (*Contreras-Moreira and Vinuesa, 2013*) and blastn (v2.11.0+) (*Altschul et al., 1990*). Orthoclusters for which pbmm2 did not align any CCS reads to the representative sequence with at least 95% identity were excluded from further analyses. Further details are provided in Appendix 1.

## Modeling

To estimate the full size of each population's NLR repertoire, we calculated the increment in the total number of identified NLR exon sequences when adding sequence data from one additional individual of a population to a set of already surveyed individuals. As noted earlier (*Medini et al., 2020*), these increments are well approximated by a power-law decay.

Briefly, given a sample of $n$ individuals, there are

$$w_n(x) = \binom{n}{x-1}(n - (x-1)) = \binom{n}{x}x \tag{1}$$

ways to choose $x-1$ individuals from the entire sample and add another – not yet chosen – one. For each $x$, we calculated the increment in the number of identified exon sequences and averaged over all possible choices of individuals. Summation of the average increments yields the total number of exons identified with $x$ individuals, as plotted in *Figure 3D*. Then, we fitted the nonlinear function

$$y = \alpha H(x, \beta) \tag{2}$$

where $H(x, \beta)$ is the generalized harmonic number with parameter $\beta$, that is,

$$H(x, \beta) = \sum_{k=1}^{x} \frac{1}{k^{\beta}} \tag{3}$$

It represents the sum of increments, decaying according to a power-law, with parameters $\alpha$ (intercept) and $\beta$ (decay rate). Importantly, if $\beta > 1$, the series in *Equation 3* converges and its limit may be interpreted as the size of a *closed* NLRome. The NLRome is *open*, if $\beta \leq 1$. Values of the fitted parameters and saturation limits are presented in *Table 1*.

## Genetic diversity

Single-nucleotide genotypes in each fish were identified from the.bam output of pbmm2 by using deepvariant (r1.0) (*Poplin et al., 2018*) with the PacBio model. Joint genotyping of the individual samples was done with glnexus (v1.2.7–0-g0e74fc4) (*Yun et al., 2021*) with its deepvariant-specific setting. Per-site $\theta_{\pi}$ of the NLR exons was calculated with vcftools (v0.1.16) (*Danecek et al., 2011*). Watterson's estimator of the scaled mutation rate is

$$\theta_w = \frac{S}{H(n-1, 1)\, l} \tag{4}$$

where $S$ is the number of segregating sides seen in a sample of $n$ aligned sequences, each of size $l$ (here, 1761 bp for the FISNA-NACHT exons and 540 bp for the B30.2 exons).

Under neutrality (all alleles confer the same fitness to an individual) and constant population size over time, one expects equality $\theta_{\pi} = \theta_w$.

## Data visualization

Plots and heat maps were created in RStudio (v2022.07.2) with R (v4.2.1) using ggplot2 (v3.3.6) or xmgrace (v5.1.25; https://plasma-gate.weizmann.ac.il/Grace/). Venn diagrams were created via

BioVenn (*Hulsen et al., 2008*; *Figure 3B*) and ggvenn (v0.1.9) (*Figure 1A*). Final processing of the images was done in Inkscape (v1.1.2; https://inkscape.org/).

## Acknowledgements

The authors are grateful to Emilia Martins and Anuradha Bhat for their contributions in wild sample collection, and to Cornelia Stein and the laboratory of Matthias Hammerschmidt for laboratory samples. We thank Bruno Hüttel and Max Planck-Genome-Centre Cologne for all the advice with library construction and for sequencing. We are also thankful for the contributions of Philipp Schiffer (help with writing the initial project proposal), Lisa Vogelsang (assistance with laboratory work), and Robert Fürst and Anna Rottmann (management of computational infrastructure). We further thank anonymous peer-reviewers for constructive feedback which helped us improve the manuscript. This work was funded by a grant to TW and ML in the frame of the priority program SPP-1819 of the German Research Foundation (DFG). JS was additionally supported by a National Sciences and Engineering Council of Canada (NSERC) Discovery Grant to Colin Garroway.

## Additional information

### Funding

| Funder | Grant reference number | Author |
| --- | --- | --- |
| Deutsche Forschungsgemeinschaft | SPP1819 | Maria Leptin<br>Thomas Wiehe |

The funders had no role in study design, data collection and interpretation, or the decision to submit the work for publication.

### Author contributions

Yannick Schäfer, Data curation, Software, Formal analysis, Validation, Investigation, Visualization, Methodology, Writing – original draft, Writing – review and editing; Katja Palitzsch, Investigation, Methodology, Writing – review and editing; Maria Leptin, Conceptualization, Resources, Supervision, Funding acquisition, Validation, Project administration, Writing – review and editing; Andrew R Whiteley, Resources, Investigation, Writing – review and editing; Thomas Wiehe, Conceptualization, Resources, Formal analysis, Supervision, Funding acquisition, Visualization, Methodology, Project administration, Writing – review and editing; Jaanus Suurväli, Data curation, Software, Formal analysis, Supervision, Investigation, Visualization, Methodology, Writing – original draft, Writing – review and editing

### Author ORCIDs

Yannick Schäfer ⓘ http://orcid.org/0000-0002-5264-8816
Katja Palitzsch ⓘ http://orcid.org/0000-0002-6292-4925
Maria Leptin ⓘ http://orcid.org/0000-0001-7097-348X
Andrew R Whiteley ⓘ http://orcid.org/0000-0002-8159-6381
Thomas Wiehe ⓘ http://orcid.org/0000-0002-8932-2772
Jaanus Suurväli ⓘ http://orcid.org/0000-0003-0133-7011

### Decision letter and Author response

Decision letter https://doi.org/10.7554/eLife.98058.sa1
Author response https://doi.org/10.7554/eLife.98058.sa2

## Additional files

### Supplementary files

• MDAR checklist

## Data availability

NLR reads are available in the NCBI Sequence Read Archive (BioProject PRJNA966920). Scripts are available on GitHub (https://github.com/YSchaefer/pacbio_zebrafish, copy archived at *Schaefer, 2024*). Sequences of the hybridization baits are provided as a source dataset.

The following dataset was generated:

| Author(s) | Year | Dataset title | Dataset URL | Database and Identifier |
|---|---|---|---|---|
| University of Cologne, Yannick Schaefer | 2023 | Targeted PacBio Sequencing of Zebrafish NLR Exons | https://www.ncbi.nlm.nih.gov/bioproject/?term=PRJNA966920 | NCBI BioProject, PRJNA966920 |

The following previously published dataset was used:

| Author(s) | Year | Dataset title | Dataset URL | Database and Identifier |
|---|---|---|---|---|
| Genome Reference Consortium | 2017 | Genome assembly GRCz11 | https://www.ncbi.nlm.nih.gov/datasets/genome/GCF_000002035.6/ | NCBI Assembly, GCF_000002035.6 |

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

# Appendix 1

## Supplementary methods

### DNA extraction

High molecular weight (HMW) DNA from laboratory zebrafish was extracted from caudal fin clips using the QIAGEN MagAttract HMW DNA extraction kit. HMW DNA from wild zebrafish was extracted from caudal fin clips using the NucleoSpin Tissue Kit from MACHEREY-NAGEL with the following adjustments. Tissues other than muscle were removed before DNA extraction with forceps. The incubation time of the Proteinase K treatment was changed from 1 to 3 hr to 10–15 min. An RNAse A treatment step was included by incubating with 400 µg RNAse A (Sigma-Aldrich) for 2 min at room temperature. All DNA samples were quantified and quality checked with Qubit 3.0 (Thermo Fisher Scientific), 0.8% agarose gels, and the 4200 TapeStation Electrophoresis System (Agilent Technologies). DNA extraction failed for one of the 20 CHT samples, but was successful for the other 95 fin clips.

### Shearing and barcoding

HMW DNA was sheared into 1.5–6 kb fragments with the red miniTUBEs of the Covaris M220 ultrasonicator. Quality control after shearing was performed using the 4200 TapeStation Electrophoresis System (Agilent Technologies). The obtained DNA fragments were size selected with 0.4× AMPure XP beads (Beckmann Coulter Inc) to exclude fragments smaller than 1.5 kb. For wild zebrafish samples, a DNA damage repair step was included in order to repair any possible DNA damage resulting from long periods of storage (particularly important for the older CHT samples). The repair step was carried out with PreCR Repair Mix (New England Biolabs).

DNA fragments were barcoded with the NEBNext Ultra II DNA Library Prep Kit (New England Biolabs) and NEBNext Multiplex Oligos for Illumina, Index Primers Set 1 (New England Biolabs). The manufacturer's standard protocol was followed until the amplification step for the enrichment of barcode-ligated fragments. At this stage, the recommended amplification protocol (PCR program) was modified to suit large DNA fragments (*Appendix 1—table 2*) and the high-fidelity Kapa polymerase (Kapa HiFi Hotstart Readymix, Kapa Biosystems) was used. The resulting barcoded DNA was purified and size-selected two more times, first with 0.5× AMPure XP beads and then with 0.4× AMPure XP beads. The amount of DNA was quantified with Qubit and quality checked with gel electrophoresis on a 0.8% agarose gel. The samples were then pooled with each pooled sample containing barcoded DNA of either four fish (CGN, TU, first library of each wild population) or eight fish (the remaining libraries of the wild populations).

### NLR capture with hybridization baits

Target enrichment was carried out according to MYbaits manual version 3.02 by using the MYbaits customized target enrichment kit for Next Generation Sequencing (MYcroarray, now part of Daicel Arbor Biosciences). The bait set contained nearly 20,000 unique 120 bp biotinylated RNA molecules in equimolar amounts. Most of the baits were designed to specifically bind to the FISNA-NACHT and B30.2 exons in the genome, but we also targeted other genes of interest. Bait hybridization and target enrichment for each pooled sample were performed according to the MYbaits manual version 3.02, with half the amount of baits and reagents used for the four-fish pools than for the eight-fish pools. Following an overnight incubation of the pooled DNA samples with RNA baits, bait-bound DNA fragments were extracted from the solution with DB MyOne Streptavidin C1 beads (Thermo Fisher Scientific). The enriched libraries were subsequently amplified with P5 and P7 primers (Illumina) by running 26 cycles of the program described in *Appendix 1—table 2*. If the DNA yield was less than 1000 ng afterward (measured by Qubit), five more PCR cycles were added. Enrichment success was evaluated by qPCR, using 5× HOT FIREPol EvaGreen qPCR Mix Plus (ROX) (Solis BioDyne) and primers specific for the FISNA-NACHT exons from each of the four groups of NLRs *Appendix 1—table 3 and 4*. The gene il1 was used as a single copy control. All primers were custom-ordered from biomers.net GmbH. The qPCR experiment was deemed successful if a strong enrichment could be seen for all NLR groups, weaker enrichment for il1, and no enrichment for the random intron. After this, the sample was selected for subsequent PacBio library construction and purified with 0.7× Ampure PB Beads (Pacific Biosciences).

## Library construction and sequencing

The final libraries were prepared with the SMRTbell Template Prep Kit 1.0-SPv3 (Pacific Biosciences). At the ligation step, the recommended amount of PacBio adapters was increased from 1 to 5 µl per 40 µl total reaction volume and the reaction was incubated overnight at room temperature. For the SN and CHT libraries in pools of eight (see *Appendix 1—table 1*), barcoded PacBio adapters were used instead of regular ones. The product codes for barcodes were BC1001 and BC1002 for CHT, BC 1003 and BC1004 for SN.

The first libraries (TU, CGN, 4 DP and 4 KG samples) were size selected to 2–8 kb with the BluePippin pulsed field electrophoresis system (Sage Science). The following libraries were size selected to 1.5–8 kb.

All sequencing was done at the Max Planck-Genome-Centre Cologne. All TU, CGN, DP, and KG zebrafish, as well as four CHT and four SN samples were sequenced with 1M v2 SMRT Cells of the Sequel instrument (Pacific Biosciences). The rest of the samples (all with barcoded adapters) were multiplexed together and sequenced with an 8M SMRT Cell of the much higher throughput Sequel 2 instrument (Pacific Biosciences). One of the already sequenced SN samples (SN24) was also resequenced in this run as it yielded no data in the first one. Furthermore, Pacific Biosciences upgraded their kits with a superior polymerase after we had sequenced TU, CGN and the first four samples of each wild population; all samples other than those were sequenced with their LR (long run) polymerase.

An overview of the sequencing is presented in *Appendix 1—table 1*.

## Read processing and assembly

Raw data were de-multiplexed and stripped of primer/adapter sequences with lima from Pacific Biosciences. For the samples sequenced with the PacBio Sequel I, the parameters –enforce-first-barcode –split-bam-named –W 100 were used with lima v1.0.0 for the runs without the LR polymerase. For Sequel runs with the LR polymerase, lima v1.8.0 and v1.9.0 were used with the same parameters. To remove PacBio barcodes from the data produced on Sequel II, lima v1.11.0 was used with parameters –split-bam-named –peek-guess and for the subsequent removal of NEBNext barcodes, the parameters were changed to –enforce-first-barcode –split-bam-named –peek-guess. Consensus sequences of all DNA fragments with a minimum of three passes (henceforth referred to as CCS reads) were calculated using ccs (v4.2.0, Pacific Biosciences) with default parameters. PCR duplicates were identified and flagged with pbmarkdup (v1.0.0, Pacific Biosciences) with default parameters, then excluded from downstream analyses. Any chimeric reads containing a primer sequence in the middle were identified with blastn (v2.11.0+) (*Altschul et al., 1990*) and removed. The filtered CCS reads were assembled into contigs for each fish separately using hifiasm (v0.15.4-r347) (*Cheng et al., 2021*) with default parameters.

## NLR identification

To obtain a list of NLR gene positions in the reference genome, we first extracted known NLR locations from Ensembl. In addition, the reference genome was translated in all frames using transeq (from EMBOSS:6.6.0.0) (*Rice et al., 2000*) and searched for further NLRs using hmmsearch from hmmer (v3.2.1) (*Wheeler and Eddy, 2013*), without bias correction and with the hidden Markov model (HMM) profiles for zf_FISNA-NACHT and zf_B30.2 from *Adrian-Kalchhauser et al., 2020*. Each position in which the zf_FISNA-NACHT model found a hit with a maximum i-Evalue of $1e-200$ and a minimum alignment length of 500 aa was considered a FISNA-NACHT exon. The filtering thresholds for B30.2 exons were an i-Evalue of $1e-5$ and a minimum alignment length of 150 aa. This approach was used both during bait design and as a preparatory step for the first round of read filtering.

To distinguish CCS reads of NLR genes from other CCS reads, the CCS reads of each fish were mapped against the reference genome GRCz11 using pbmm2 (v1.3.0) with preset ccs. CCS reads which mapped within a known NLR gene or one found with our HMM-based approach with any mapping quality were considered potential NLR reads and used as input for subsequent steps.

De novo assembled contigs containing NLR exon sequences were identified by translating all contigs of each fish in all frames with transeq (from EMBOSS:6.6.0.0) and subsequently searching for FISNA-NACHT and B30.2 domains using hmmsearch from hmmer (v3.2.1) without bias correction and the HMMs zf_FISNA-NACHT and zf_B30.2 again. The HMM-based approach was chosen for the contigs in particular because we assumed that there would be NLR sequences in the data which

are absent in the reference genome and therefore might not be mapped. The approach enabled us to include all FISNA-NACHT and B30.2 exon data found by *Adrian-Kalchhauser et al., 2020* in our searches, optimizing the search sensitivity.

By examining all NLRs annotated in the reference genome, we found a highly conserved 47 bp exon preceding B30.2 to be present in most NLR-B30.2 genes (NLRs containing a B30.2 domain), but not in other NLRs nor in most other B30.2-containing genes. B30.2 exons from NLRs were distinguished from B30.2 elsewhere in the genome by generating a HMM for the 47 bp exon based on the blast hits and searching the contigs for matches to this model with hmmsearch from hmmer (v3.2.1). The model was created with hmmbuild from hmmer (v3.2.1).

The FISNA-NACHT and B30.2 orthoclusters were postprocessed after get_homologues as follows: whenever an orthocluster contained more than one contig, a consensus sequence for the cluster was created from all those contigs with cons from EMBOSS:6.6.0.0. These consensus sequences and the contig sequences of the singleton clusters made up the representative sequences of the orthoclusters. Some representative sequences were reversed with revseq from EMBOSS:6.6.0.0 so that all exons were in the same orientation. The representative sequences were then blasted against each other using blastn (v2.12.0+) with default parameters and output format 6. In cases in which 98% and at least 3 kb of a representative sequence matched another with at least 98% identity, the two clusters they represented were fused into a new cluster by combining their contigs and generating a new consensus sequence from them. This process was conducted twice and reduced the number of FISNA-NACHT clusters from the initial 4743 to 2008 and B30.2 clusters from 14,879 to 2,635.

The bam files produced by mapping the NLR reads of each fish separately to the representative sequences of the orthoclusters were filtered using samtools (v1.7) (*Li et al., 2009*). If the representative sequence had at least one primary alignment (SAM flag 0 or 16) with length >1 kb, mapping quality 60, and no more than nine soft-clipping bases at both ends of the mapped read, the orthocluster was assumed to occur in the respective fish.

Circular genome plots were created with circos (v 0.69–8) (*Krzywinski et al., 2009*) running on Perl 5.036000. Principal component analysis of scaled NLR counts per individual was conducted and plotted with the R packages ade4 (v1.7-22) and adegraphics (v1.0–21) (*Thioulouse et al., 2018*).

**Appendix 1—table 1.** Sequencing scheme for the zebrafish samples.
Libraries sequenced after the introduction of an improved (long run) sequencing chemistry are marked with LR. Samples that yielded no data after sequencing are marked with asterisks.

| Individuals | Library | Sequencer |
| --- | --- | --- |
| TU01, TU02, TU03, TU06 | TU L1 | Sequel |
| TU08, TU10, TU12, TU14 | TU L2 | Sequel |
| CGN1, CGN2, CGN3, CGN4 | CGN L1 | Sequel |
| CGN5, CGN6, CGN7, CGN8 | CGN L2 | Sequel |
| DP07, DP09, DP10, DP12 | DP L1 | Sequel |
| DP15, DP20, DP23, DP24, DP25, DP28, DP31, DP34 | DP L2 | Sequel (LR) |
| DP03, DP05, DP13, DP16, DP21, DP29, DP31, DP33 | DP L3 | Sequel (LR) |
| KG35, KG41, KG42, KG43 | KG L1 | Sequel |
| KG03, KG05, KG07, KG12, KG14, KG15, KG18, KG19 | KG L2 | Sequel (LR) |
| KG20, KG22, KG24, KG26, KG29, KG32, KG33, KG44 | KG L3 | Sequel (LR) |
| SN21, SN23, (SN24*), SN26 | SN L1 | Sequel |
| SN03, SN04, SN08, SN09, SN10, SN11, SN12, SN24 | SN L2 | Sequel II (LR) |
| SN13, SN14, SN15, SN16, SN17, SN18, SN19, SN20 | SN L3 | Sequel II (LR) |
| CHT19, CHT23, CHT26, CHT28 | CHT L1 | Sequel |
| CHT01 - CHT07, (CHT13*) | CHT L2 | Sequel II (LR) |

*Appendix 1—table 1 Continued on next page*

*Appendix 1—table 1 Continued*

| Individuals | Library | Sequencer |
|---|---|---|
| CHT08, CHT10 - CHT12, CHT14 - CHT16, (SN25*) | CHT L3 | Sequel II (LR) |

**Appendix 1—table 2.** PCR program used for barcoding.
For library amplification, the same program was used with 26 or 31 cycles.

| Step | Temperature (°C) | Duration | |
|---|---|---|---|
| Initialization | 98 | 4 min | |
| Denaturation | 98 | 30 s | |
| Annealing | 65 | 30 s | {x 12} |
| Elongation | 72 | 12 min | |
| Final elongation | 72 | 20 min | |
| Storage | 4 | ∞ | |

**Appendix 1—table 3.** qPCR program for the evaluation of enrichment efficiency.

| Step | Temperature (°C) | Duration | |
|---|---|---|---|
| Initialization | 95 | 12 min | |
| Denaturation | 95 | 15 s | |
| Annealing | 65 | 20 s | {x 40} |
| Elongation | 72 | 20 s | |

**Appendix 1—table 4.** Sequences of qPCR primers used for evaluation of target enrichment.

| Gene | Direction | Sequence |
|---|---|---|
| il1 | + | 5'-tgg-tga-acg-tca-tca-tcg-cc-3' |
| il1 | - | 5'-tcc-agc-acc-tct-ttt-tct-cca-a-3' |
| foxo6 intron | + | 5'-agt-tct-gtg-tgg-gaa-cag-gg-3' |
| foxo6 intron | - | 5'-gtg-cat-ctt-tag-cgt-tgg-ct-3' |
| NLR group 1 | + | 5'-cct-gac-aca-ggt-caa-caa-aac-a-3' |
| NLR group 1 | - | 5'-gat-tgt-ctt-ttc-ctt-cag-ccc-ag-3' |
| NLR group 2 | + | 5'-tgg-att-ggg-ctg-aag-gga-aa-3' |
| NLR group 2 | - | 5'-agg-ttc-agt-cct-tta-gtc-tct-gg-3' |
| NLR group 3 | + | 5'-ctg-ctg-gag-gtg-aaa-gat-cag-ac-3' |
| NLR group 3 | - | 5'-gat-tgt-tga-gca-gtg-agc-agg-a-3' |
| NLR group 4 | + | 5'-tac-ctg-gac-aag-aca-aag-cca-3' |
| NLR group 4 | - | 5'-ctc-ctt-ctc-ttc-agc-cca-gtc-3' |

