## [Editor Report]

This useful study employs a sequence capture approach to characterize the diversity of NLR sequences in wild zebrafish populations. The authors provide solid evidence that wild zebrafish populations harbor several thousand NLR genes in total, with individual fish having a few hundred NLR gene copies.

---

## [Decision Letter]

[Editors' note: this paper was reviewed by Review Commons.]

---

## [Author Response]

1. General Statements

We thank the reviewers for their constructive feedback on our study. We have updated the manuscript according to their recommendations.

Two of the three reviewers raised concerns about the quality of some of our data, in particular of the DP population. Since most DP samples suffered from low coverage, introducing bias when being compared to the higher quality data, we moved all results involving DP to the Supplement. We mention this in the Discussion. the main text and its figures were updated to focus on the remaining three wild populations and two laboratory strains.

We rewrote parts of the Abstract, Introduction and Discussion to clarify the comparison between the NLR families of *Arabidopsis thaliana* and *Danio rerio*. In regard to this, we also added an extra panel to figure 3.

2. Point-by-point description of the revisionsReviewer #1 (Evidence, reproducibility and clarity (Required)):General comments:1) Odd statement in abstract comparing plants to fishes – why not put in context of animals in general, or vertebrates in particular? Same in the introduction and discussion. Either a more broad comparative framework should be set – from plants, through invertebrates to vertebrates, or it should be shortened and perhaps discussed in the light of diversity present in e.g., fish or just vertebrates. A persistent focus on plants is peculiar and does not seem to provide adequate background/context.

We have made an effort to clarify the reason for mentioning NLRs from *A. thaliana* in the Abstract, Introduction and Discussion. We added a plot to figure 3, which is analogous to figure 3A from Van de Weyer et al., 2019 where the pan-NLRome of *A. thaliana* is analysed.

Furthermore, we added a paragraph discussing copy number variation of another immune gene family (MHC genes) in vertebrates.

2) Methods seem overall well described, and the analyses seem to have been performed diligently-although I am not an expert in the field. One aspect that would be important to evaluate methodology is the repeatability of NLR detection – If we perform the procedure twice on DNA sample from a single individual, how repeatable collection do we get?

It is possible that we might have failed to identify a few NLRs from some individuals. Unfortunately, sequencing could not be repeated due to the limited amount of raw tissue. the population DP in particular is problematic since we had low sequencing depths across all of its samples. We have now excluded it from the main results and moved it to the Supplement. Still, we are confident that the general feature of widespread copy number variation both within and between populations persists even if the experiments were repeated. One argument for this is that the non-linear functions describing pan-NLRome sizes are fairly consistent among wild populations. Also, sample sizes of about 20 individuals per (wild) population are large enough to compensate for poor data quality of one or two individuals. Finally, we found a substantial fraction of NLRs to be present in all wild populations, including DP. It would be unlikely to detect so many shared genes if the amplified sequences were spurious or the NLR identification procedure faulty.

We have expanded the discussion of the caveats of our approach.

3) Discussion is quite speculative, and some claims seem exaggerated; in conclusions, eg: "This study advances our understanding of the evolutionary dynamics affecting very large gene families." – as the study is mostly descriptive documentation od CNV of gene (or, gene fragments) I am not convinces how it really advances "understanding of the evolutionary dynamics".

We toned down Discussion and Conclusions.

Minor points:At this stage the journal and so the audience is unknown, so perhaps this will not be an issue – but for a broader audience, a better explanation of what PRY/SPRY/B30.2 are would be useful.

We added an explanation.

English could be smoothed, certain sentences sound odd: eg. lines 87-89; or 90-93 – Studies have shown that viral and bacterial infections can induce the expression of specific fish NLRs (reviewed in (24)). Some have PYD or CARD domains and can even form inflammasomes similar to mammalian NLRs (25, 26). – could be read as if infections had PYD or CARD domains, not NLRs. Lines 239-240 – not sure what does "presence/absence variation" mean here.We made an effort to improve language style.Sequence of baits used should be provided in some supplement or repository.

The sequences are now attached to the manuscript as a supplementary dataset.

Reviewer #1 (Significance (Required)):The article tackles an overall interesting subject: an expansion of a relevant group of genes in one of the major model species of biomedical importance. the main strength of the study is putting in context variation found in the lab strains – via comparison to wild populations. A limited and "haphazard" genetic variation (especially of genes involved in immune processes) of laboratory models can have paramount implications for the interpretation of experimental studies.

We expand on this in the Discussion.

That being said, expansion of NLR family in teleosts in general, and in zebrafish in particular, has been previously described (eg, https://bmcecolevol.biomedcentral.com/articles/10.1186/1471-2148-8-42), and this study mainly expands description of this phenomena.

The expansion of NLRs has indeed been described before and in great detail. the manuscript contains multiple references on this topic, including the one recommended by the reviewer (e.g. Stein et al., 2007, Laing et al., 2008, Howe et al., 2016). the previously reported expansion of NLRs in the reference genome was a description, but not an interpretation "in the light of evolution". We demonstrate that gene birth, death, and presence/absence variation are ongoing processes and active on a population-genetic time scale.

Given the methodology, little can be said about possible functions of the discovered diversity. In particular, a technical aspect imposing limitation of sequence length resulted in a collection of exons, but not full genes – precluding e.g., deepened analysis of domain architecture.

We agree that our study does not reveal much about the function of the NLR-C genes. the focus on function in the discussion was disproportionate to our findings and we have reduced it to avoid speculation.

Selection analysis is also quite limited in scope; there are several, more sophisticated tools to infer balancing or purifying selection, either pervasive or episodic (see eg. excellent tools of https://www.datamonkey.org/).

Yes. However, with our limited amount of data, we deliberately do not want to speculate too much about selection. As mentioned in the text, many genes are monomorphic in our sample. To reliably infer the action of selection, and more so to distinguish it from signatures of demographic history, a much broader basis of sequence data is needed.

Nonetheless, the study certainly highlights the possible importance of such an expanded group of genes in this species, with a potential to inspire further – more mechanistic/functional research in this area. the article will likely interest a few groups of rather specialized audience – e.g., those working in NLR genes in particular, or in immunity of zebrafish (or fish in general). Another, somewhat broader angle, able to attract a wider audience would be subjects of general evolution of multigenic families (birth-and-death models, trade-offs at different CNV etc.) and comparative analysis to other groups of animals – or genes – evolving in a similar way. My field of expertise is diversity and evolution of adaptive branch of the immune system in vertebrates, in particular non-model vertebrates.Reviewer #2 (Evidence, reproducibility and clarity (Required)):Summary:The authors have examined the variation of the immune related gene family "Nucleotide-binding domain Leucine-rich Repeat containing" (NLR) in the zebrafish *Danio rerio*. These proteins, while highly divergent, are conserved across animal and plant species. In humans or rodents, the number of these genes is relatively small (20-30) but other species such as Arabidopsis appear to have >10,000 NLR genes.Schäfer et al. collected 67 wild-type zebrafish from 4 independent sites near the Bay of Bengal and added 8 fish each from the laboratory strains Tübingen (TU) and Cologne (CGN). Using DNA capture and sequencing technology, they identified 1,560 unique "FISNA-NACHT" domains representing different NLR genes and 574 NLR-associated B30.2 domains. A subset, 714 and 229 respectively, were identified in all of the fish populations representing a "core" set common NLR genes. A significant subset of the genes could not be aligned to the GRCz11 reference suggesting significant variation across subpopulations of zebrafish.Major comments:None. the work is straight-forward, carefully done and conservatively interpreted.Minor comments:GRCz11 is still a very fragmented assembly, particularly chr4 where heterochromatin repeats on the long arm were intractable to short read sequences. There is a more recent assembly generated by the Tree of Life initiative (GCA_944039275.1) from the SAT fish line that may allow a more robust alignment and placement of the NLR genes, perhaps even some phasing of unassembled fragments.

We have examined some of the existing long-read based zebrafish assemblies and found that even the length of chromosome 4 that contains most of the NLRs can significantly differ between different strains and different genome assemblies. This is also in agreement with the recent findings of McConnell et al., 2023 on large structural differences on this chromosome between three strains derived from the AB genetic background. We now mention this finding and the alternative existing zebrafish assemblies in the Discussion.

Reviewer #2 (Significance (Required)):General assessment:Obtaining more data on the inherent variation within species of NLR genes as well as collecting more cross-species data for evolutionary comparisons is valuable for our understanding of this interesting but poorly understood class of immune response genes as well as the dynamics of gene duplication/deletion in complex, repeated arrays.-The major strength of the study is the efforts put into capturing wild-type zebrafish from multiple different locations to maximize the diversity of the NLR sequences.-The primary limitation is targeted capture is always contingent on having enough homology in the probes to capture all the desired genes in roughly even proportions. Some of the more interesting NLR genes might have diverged too much to be properly captured but could provide critical information on evolutionary functional adaptation. Similarly, while based on cost, it is understandable why sequence capture was chosen, but phasing information across clusters could help explain (or discover) very diverse haplotype sequences in the introns of many of these gene arrays. As sequencing costs drop, this is an important aspect to the evolution of zebrafish chromosome 4 (as well as the rest of the genome). "Pangenome" differences in the zebrafish genome might be quite radically different.

Yes, we are aware of this possibility and discussed it now in more detail.

Advance:The study represents an important characterization of a poorly understood but important class of immune genes. A very similar (but more detailed) characterization of Arabidopsis NLR genes by Van de Weyer et al., Cell 2019, is considered an important advance to the plant biology community and has been highly cited since its publication. the collected data is potentially quite useful for future studies that want to understand the complex dynamics of highly repetitive, yet functional regions of the genome and how it can result in the creation and extinction of new genes.Audience:There are two main likely audiences, those interested in how NLR genes are involved immune protection and those interested in genomes and evolution. It probably doesn't reveal a fundamental biological principle that would be of broad, general interest to the research community.Reviewer #3 (Evidence, reproducibility and clarity (Required)):Summary:The authors performed exon capture sequencing of FISNACHT and B30.2 domains from individuals among various inbred and wild zebrafish populations. This study provides average numbers of domains per fish and thus a valuable estimate for the size of the pan-NLRome in zebrafish.Major comments:- Are the key conclusions convincing?

Yes, and overall their findings are also consistent with what has been found in other model systems, most notably Arabidopsis (Weyer 2019, https://www.cell.com/cell/fulltext/S0092-8674(19)30837-2).

- Should the authors qualify some of their claims as preliminary or speculative, or remove them altogether?'Complex patterns of inheritance' is unclear from significance statement.

We have reformulated this sentence.

For mechanisms driving copy number differences, haplotypes/segregation need not be presented as a model opposed to birth and death gene evolution/tandem duplication. These may both operate at different scales.

Yes, we reformulated.

- Are the data and the methods presented in such a way that they can be reproduced?Yes, overall, though the authors could provide sequences for their adapters, similar to Weyer 2019. It seems that raw data for this study is not yet available in NCBI: https://www.ncbi.nlm.nih.gov/bioproject/966920

The data will be made available at the time of publication. the sequences for all baits used, including non-NLR-ones, have been added as a supplementary.fasta-formatted text file.

Minor comments:- Specific experimental issues that are easily addressable.Their targeted exon capture approach cannot ensure that all NLR genes will be sequenced. This method also does not sequence across other exons from NLR genes, providing only a partial view of NLR gene structure and evolution. This may miss for example additional integrated domain architectures or evidence of physical clustering in the genome (Thatcher 2023, https://bsppjournals.onlinelibrary.wiley.com/doi/10.1111/mpp.13319). Nevertheless, focusing on capturing and enumerating FISNACHT domains as the most conserved and characteristic domain for NLRs appears to be a reasonable approach for estimating rough copy numbers.

We agree with the reviewer and have now added explanatory text on this topic to the discussion.

Unfortunately, mapping these domains back to only a single reference genome also means that the accuracy of their predictions for these domains, including which belong to genes or pseudogenes, is likely to be more error-prone (Wang 2019, https://www.pnas.org/doi/10.1073/pnas.1910229116), particularly for more divergent sequences. This may lead to additional limitations for assigning one-to-one relationships between FISNACHT domains, which they measure, and genes, which they seek to enumerate.

Thank you for pointing out the above reference. This is a limitation that we realized early on in the study after which we decided to not rely on the reference genome at all, instead opting to assemble and cluster all data de novo. In the manuscript we only use the reference genome for checking which of our identified NLRs have a clear homologue to satisfy potential interest by the biomedical community. All the other analyses were conducted based on the de novo assemblies and orthologous clusters.

In addition, many of their samples appear to be of poor DNA quality, meaning thatmodeling estimates of domain copy number for the population can appear erroneous, even perhaps off by as much as a factor of 10 for the DP strain. This may contribute to some curious 'artifact' in some of the figures, such as FiguresS2B1, S3B, S4C2, particularly for the B30.2 sequences in DP. the authors mention 'low sequencing depths' and 'low coverage' as contributing factors for this 'artifact' but also invoke possible 'evolutionary factors' in the discussion. Additional discussion for the apparent experimentally-induced effects on underestimating B30.2 sequences (perhaps due to increased DNA fragmentation and smaller domain size?), rather than coincidence of low coverage and actual biological strain-specific loss of B30.2 domains, appears warranted.

As mentioned in the text, although samples from the population CHT were older than the others and slightly degraded, they were rescued by applying the PreCR Repair Mix from New England Biolabs. We incorporated incubation with the reagent to the standard protocol that we used for all samples. the other samples had no quality problem.

Furthermore, the reviewer raises concerns regarding data quality for the DP population samples. After careful consideration, we decided to exclude the DP data and its analysis from the main text and deferred it to the supplementary material with the necessary remarks of caution. the main text now focuses on the other three wild populations, which did not have those issues.

Additionally, as per the reviewer’s suggestion, we discuss the additional caveats of bait-based targeted sequencing, especially for B30.2 domains.

- Do you have suggestions that would help the authors improve the presentation of their data and conclusions?Statement in the abstract about fewer (not 'less') NLRs in lab strains vs wild: consider revising to average # NLRs per individual.

We changed the respective sentence in the abstract.

Revise Figure 3C legend: 'Totally discovered NLR genes', e.g., to 'total # NLR genes'.

The recommended change was made to the figure legend.

Reviewer #3 (Significance (Required)):- Describe the nature and significance of the advance (e.g. conceptual, technical, clinical) for the field.This study elucidates the pan-NLRome of zebrafish via application of a somewhat new technique.- Place the work in the context of the existing literature (provide references, where appropriate).The authors apply a targeted capture and long read technology previously used in plants (Weyer 2019) to fish, thereby elucidating the pan-NLRome of zebrafish, a vertebrate with a large # of NLR genes. Previous studies has shown NLR gene variation in zebrafish, but had not compared levels within and between strains, or estimated a total NLR # across populations.- State what audience might be interested in and influenced by the reported findings. Those interested in immune function, genetic diversity and genome evolution.- Define your field of expertise with a few keywords to help the authors contextualize your point of view. Indicate if there are any parts of the paper that you do not have sufficient expertise to evaluate.Immune genes and evolution. the specific statistical approach